# OpenReview forum: "Auto-Enhance: Towards a Meta-Benchmark to Evaluate AI Agents' Ability to Improve Other Agents"
_NeurIPS.cc/2024/Workshop/SafeGenAi — SafeGenAi Poster_

### Official Review · Reviewer_XNmt · 2024-10-08

**Rating:** 7
**Confidence:** 3

**Review:**

The paper presents a novel "meta-benchmark" problem where a top-level agent is tasked to improve the performance of some reference agents on some reference benchmarks. The paper presents a thorough evaluations across multiple challenging LLM agent benchmarks such as in software engineering. The work would inspire more multi-agent collaboration evaluation methods. One of the weaknesses that are not mentioned is the variances and statistics of these improved performance and how they can be quantified in a step-by-step manner.

---

### Official Review · Reviewer_dM88 · 2024-10-10
**The manuscript is ok but not good enough**

**Rating:** 4
**Confidence:** 4

**Review:**

Pros: This paper leveraged AI agents to focus on the research creating general, extensible and standardised measurements of self-improvement capabilities. A meta-benchmark for evaluating how LLM agents can improve other LLM agents across a range of simple and complex real-world tasks.

Cons:
1. Figure 1: Progress against milestones reached by ReAct agents, with and without an SSH-tool, on variants of four enhancement tasks from our benchmark. In two different conditions, whether the results obtained by the benchmark test are significantly different, it is suggested to add relevant content.
2. Line 132: Agents did not make significant progress on the hard variant, often hallucinating flags, commands, or files. The reason for this sudden occurrence of instructions is based on what parameters are relevant to the LLM agents or otherwise.
3. Line 144 and 145: However, the agent uses a non-existent column of the datasets and uses activations from the last layer, which won’t work and needs "hyperparameter-ization". In addition to the above mentioned conditions, when the agent performs the basic run operation, what other conditions do not take effect.
4. Line 165: We find that an agent fixes both errors within 25 steps with this setup.Can a strategy be proposed to improve the speed at which the agent fixes errors and the efficiency at identifying errors?
5. Line 181-183: Future work to expand the library of included tasks, and ideally to incorporate benchmarks wholesale, would allow results to become more granular, less arbitrary, and more robust and likely to generalise. How to achieve the ideal expression of results, and what specific changes can be made in the research basis of this paper?

---

### Official Review · Reviewer_wEny · 2024-10-12
**a valuable initial step toward understanding AI self-improvement**

**Rating:** 6
**Confidence:** 4

**Review:**

### Strengths
- a meta-benchmark to assess agents' self-improvement is timely and innovative.
- the use of multiple domains (software engineering, cybersecurity, ML experimentation) strengthens this evaluation framework.
---
### Weaknesses
- manually defined milestones for evaluation introduces subjectivity
- the experiments show significant fragility and limitations in planning and execution by agents, which weakens the overall impact.
---
### Suggestions
- It would be better to try to explore strategies and possible ideas to mitigate the frequent failure modes, like better task decomposition or improved planning tools.